# Preparation and Evaluation of Rebamipide Colloidal Nanoparticles Obtained by Cogrinding in Ternary Ground Mixtures

**Yayoi Kawano** [1] , **Yuiko Utsunomiya** [1] , **Fumiya Yokoyama** [1] , **Naoko Ishii** [1,2] and **Takehisa Hanawa** [1,*]

1   Faculty of Pharmaceutical Sciences, Tokyo University of Science, 2641 Yamazaki, Noda, Chiba 278-8510, Japan; y.kawano@rs.tus.ac.jp (Y.K.); 3b18509@alumni.tus.ac.jp (Y.U.); 3a13099@alumni.tus.ac.jp (F.Y.); ishii@kashiwacity-hp.or.jp (N.I.)
2   Department of Pharmacy, Kashiwa City Hospital, 1–3 Fuse, Kashiwa, Chiba 277-0825, Japan
*   Correspondence: t-hanawa@rs.tus.ac.jp; Tel.: +81-4-7121-3654

**Abstract:** Aphthous stomatitis is one of the side effects of chemotherapy and radiotherapy in cancer treatment. Rebamipide (RB) mouthwash for stomatitis acts as a radical scavenger. However, RB is poorly soluble in water, which leads to aggregation and precipitation of the dispersoid. The particle size of the drug needs to be less than 100 nm for the particles to reach the mucus layer in the oral cavity. In this study, we attempted to prepare nanoparticles of RB by cogrinding with polyvinylpyrrolidone (PVP) or hydroxypropyl cellulose (HPC) and sodium dodecyl sulfate (SDS) using a mixer ball mill, and evaluated the physicochemical properties of RB nanoparticles, the stability of dispersion in water, and permeation of the mucus layer in vitro. By cogrinding, the particle size decreased to around 110 nm, and powder X-ray diffraction (PXRD) of the particles showed totally broad halo patterns, which suggested a decreased crystalline region. Furthermore, the solubility of RB nanoparticles increased by approximately fourfold compared with RB crystals, and the water dispersibility and permeation of the mucus layer were improved. The results suggest that in a ternary ground mixture of RB, PVP or HPC, and SDS, the RB nanoparticles obtained can be applied as a formulation for stomatitis.

**Keywords:** rebamipide; stomatitis; mouth wash; nanoparticles; ternary ground mixture; stability of dispersion; mucus permeation

## 1. Introduction

Cancer is the second leading cause of death according to estimates by the World Health Organization (WHO) in 2018 [1]. The treatments are roughly classified into surgery, chemotherapy, and radiotherapy [2,3]. However, many side effects have been reported, including cell toxicity (such as myelosuppression, cardiac toxicity, renal dysfunction, hepatic dysfunction, and stomatitis) and digestive symptoms (such as nausea and diarrhea) [4–7]. In particular, 40–70% of patients develop stomatitis, which induces pain in the oral cavity, decreases food intake, causes psychological stress, and reduces the quality of life (QOL) remarkably [8,9]. Furthermore, treatment and/or prevention of stomatitis is an important issue for chemotherapy or radiotherapy if it becomes serious [8,9].

The pathogenic mechanism of stomatitis differs between chemotherapy- and radiotherapy-induced forms. In chemotherapy, stomatitis is caused by the formation of free radicals, which cause oxidative stress in the oral mucosa. Additionally, it is caused by decreased immunity, such as bacterial infection in the oral cavity, undernutrition, and myelosuppression. On the other hand, free radicals are produced by radiation, which damages the oral mucosa. Furthermore, radiation damages the salivary gland tissue. It has been reported that stomatitis is caused by a decrease in self-cleansing action in the oral

cavity due to decreased salivation or decreased immunity [10]. The pathobiology of stomatitis is classified into five stages: phase I (initiation), phase II/III (messaging, signaling, and amplification), phase IV (ulceration (stomatitis)), and phase V (healing) [10].

Currently, indomethacin, which has an analgesic effect, or allopurinol and rebamipide (RB), which have a free radical scavenging effect, are administered to treat stomatitis caused by chemotherapy and radiotherapy [8,11–19]. Allopurinol has a good treatment effect on stomatitis; however, it interacts with fluorouracil, which decreases its anticancer effect, and it has been reported that there are restrictions to its administration [20,21]. On the other hand, RB can be administered in any case because there is no interaction with anticancer agents [22]. In addition, it has been reported that RB works effectively to treat stomatitis as a free radical scavenger, as an anti-inflammatory drug, and by inducing mucosal repair in phases I–IV [8,11,23].

RB has been reported to be effective in treating stomatitis caused by chemotherapy or radiotherapy. Actually, RB is used as a gastric mucosal protectant in Japan. However, mouthwashes containing RB have never been used as commercial products, thus RB mouthwashes for curing stomatitis have been prepared as in-hospital formulations by pharmacists at many hospitals in Japan. In these cases, to prepare a mouthwash, RB tablets are ground suspended in water. The preparation process is not difficult; however, once precipitation occurs, it is not easy for patients to redisperse the solution. The mouthwash undergoes aggregation and precipitation of the dispersoid, including RB crystals, because RB is a poorly water-soluble drug. Therefore, RB crystals in the mouthwash cannot easily reach the mucous layer because the mucin layer covers it and the RB has poor water solubility. To solve these problems, it is necessary to improve the stability of the dispersion of the mouthwash and permeation of the mucin layer by using, for example, a nanosuspension. Some methods to improve the dispersion include micronizing the RB, increasing the viscosity, and improving the positive electrification of the dispersion [24–26]. Of these methods, RB nanoparticles are the best because the particles of the drug can be delivered to the mucous layer when their size is reduced to about 100 nm [26–28]. For this reason, in this study, we attempted to prepare RB nanoparticles to improve the stability of the dispersion and permeation of mucin layers.

The field of nanotechnology with polymers is one of the most popular areas of current research not only for drug delivery, but also for device technology [29,30]. Many methods for the preparation of nanoparticles, such as polymeric nanoparticles, micelles, and liposomes, have been reported [30–42]. Basically, there are two approaches for the preparation of nanoparticles: bottom-up and top-down. These techniques are carried out to prevent precipitation or recrystallization and to reduce the particle size in mechanical processing, respectively [43]. In the bottom-up technique, a poorly water-soluble drug is dissolved in an organic solvent and subsequently precipitated by mixing with a nonsolvent. Alternatively, another technique is to regulate crystallization [43]. Then, the particles prepared by these methods must be stable. However, the crystal growth of particles after precipitation or recrystallization is one of the problems in the technique [43]. On the other hand, the top-down technique may be a more popular method due to its ability to reduce only the particle size [43]. Especially on the laboratory scale, the preparation of nanoparticles using wet milling methods is known as a simple top-down technique to improve the solubility of poorly water-soluble drugs [31–37]. The use of ternary ground mixtures with a surfactant and water-soluble polymers prepared by various vibrational ball mills has been reported as another simple technique for the preparation of nanoparticles [37,44–46]. Therefore, we focused on ternary ground mixtures using a vibrational ball mill for the preparation of RB nanoparticles in this study.

In this study, RB nanoparticles with polyvinylpyrrolidone (PVP) or hydroxypropyl cellulose (HPC) and sodium dodecyl sulfate (SDS) as a surfactant were prepared using a desktop-type mixer ball mill in order to evaluate the physicochemical properties of the mixtures and the stability of the dispersion and permeation of the mucin layer.

## 2. Materials and Methods

### 2.1. Chemical Reagents

RB was purchased from Tokyo Chemical Industry Co., Ltd. (Tokyo, Japan). HPC (HPC-L, molecular weight of about 140,000; HPC-SL, molecular weight of about 100,000; and HPC-SSL, molecular weight of about 40,000) was provided by Nippon Soda Co., Ltd. (Tokyo, Japan). PVP (PVPK-90, molecular weight of about 360,000; PVPK-30, molecular weight of about 40,000) was purchased from Nacalai Tesque Inc. (Kyoto, Japan). SDS was purchased from FUJIFILM Wako Pure Chemicals Co., Ltd. (Osaka, Japan). Mucin was purchased from Sigma-Aldrich, Inc. (St. Louis, MO, USA). Acetonitrile (HPLC grade) and distilled water (HPLC grade) were purchased from Kanto Chemical Co., Inc. (Tokyo, Japan). All chemicals and solvents were of analytical reagent grade.

RB mouthwash was prepared by the Department of Pharmacy at Kashiwa City Hospital (Kashiwa, Japan) as the reference formulation, and its composition is shown in Table 1. Six Mucosta® tablets including 100 mg as RB were ground by a tablet grinder (Iwatani Corporation, Tokyo, Japan) for 30 s. After grinding, the powder was sieved through a No. 60 mesh sieve. The obtained powder was weighed, then citric acid (Kozakai Pharmaceutical Co., Ltd., Tokyo, Japan), L-glutamine (Nipro Corporation, Osaka, Japan), and simple syrup were added as additives. Additionally, dextrin (Toromeiku clear, Meiji Co., Ltd., Tokyo, Japan) as a thickening agent and 2 mL of preservation solution containing propyl or methyl p-hydroxybenzoate (Kanto Chemical Co., Inc. Tokyo, Japan) were added to the formulation. Finally, distilled water was added for a total of 300 mL of the formulation.

**Table 1.** Composition of rebamipide (RB) mouthwash at Kashiwa City Hospital.

| Mucosta® Tablets 100 mg | 6 Tablets |
|---|---|
| Citric acid | 0.45 g |
| L-glutamine | 1.50 g |
| Simple syrup | 12.0 mL |
| Dextrin | 1.50 g |
| Preservation solution | 2.0 mL |
| Distilled water | Total of 300 mL |

### 2.2. Preparation of Samples

RB, PVP or HPC, and SDS (weight ratios of 1:1:1, 1:3:1, and 1:5:1) were weighed in a glass screw vial (10 mL) and mixed using a vortex mixer for 2 min to create physical mixtures (PMs) (Table 2).

**Table 2.** Formulation of various samples. PM, physical mixture; GM, ground mixture; HPC, hydroxypropyl cellulose; PVP, polyvinylpyrrolidone; SDS, sodium dodecyl sulfate.

| Sample | Mixing Weight Ratio | | | Grinding Time (min) |
|---|---|---|---|---|
| | RB | HPC or PVP | SDS | |
| PM (1:1:1) | | 1 | | |
| PM (1:3:1) | 1 | 3 | 1 | |
| PM (1:5:1) | | 5 | | |
| GM (1:1:1)-15 | | 1 | | |
| GM (1:3:1)-15 | 1 | 3 | 1 | 15 |
| GM (1:5:1)-15 | | 5 | | |
| GM (1:1:1)-30 | | 1 | | |
| GM (1:3:1)-30 | 1 | 3 | 1 | 30 |
| GM (1:5:1)-30 | | 5 | | |
| GM (1:1:1)-45 | | 1 | | |
| GM (1:3:1)-45 | 1 | 3 | 1 | 45 |
| GM (1:5:1)-45 | | 5 | | |

Ground mixtures (GMs) were prepared by grinding PMs using a mixer ball mill (MM400, Retsch, Haan, Germany) with a 12 mm stainless-steel ball for 15, 30, and 45 min. The grinding was performed at 30 Hz, and the stainless jars were frozen under liquid nitrogen for 5 min before grinding.

### 2.3. Physicochemical Properties of Samples

#### 2.3.1. PXRD

Powder X-ray diffraction patterns of various samples were measured using a RINT-2000 (Rigaku Co., Tokyo, Japan) under the following conditions: filter, Ni; target, Cu; voltage, 40 kV; current, 40 mA; scanning range, 5–40°; scanning speed, 4 °C/min.

#### 2.3.2. ATR-FTIR

FT-IR spectra of samples were obtained using a Frontier FT-IR spectrometer (PerkinElmer Co., Ltd., Waltham, MA, USA) by attenuated total reflection under the following conditions: spectral resolution, $4$ cm$^{-1}$; scanning range, 4000–500 cm$^{-1}$; sample thickness, 1.0 mm; accumulation count, 16. The spectra were observed using a single bounce diamond anvil attenuated total reflection (ATR) cell.

#### 2.3.3. Particle Size and Zeta Potential in Water

For the test samples, 5 mg of GM was dispersed with 10 mL of ultrapure water in a test tube, then sonicated for 1 min. The particle size and zeta potential were obtained using a particle size and zeta potential analyzer (ELSZ-2000ZS, Otsuka Electronics Co., Ltd., Osaka, Japan) at 25 °C, then evaluated by a dynamic light scattering method and Smoluchowsi's equation, respectively. The particle size was obtained at a wavelength of 660 nm at a scattering angle of 165°, and the mean after 70 accumulated times per sample (3 min data collection period) was recorded. In this study, 3 batches were prepared for measuring.

#### 2.3.4. Viscosity of Sample Solution

Various 0.2% *w/v* GM suspensions were prepared by dispersing the GM in ultrapure water and sonicating for 1 min. Viscosity was measured using a viscometer (LV DV2T, AMETEK Brookfield, Inc., Middleborough, IN, USA) under the following conditions: cone rotor, CPA-40Z; rotational frequency, 50 rpm; shear rate, 375.0 s$^{-1}$; measurement time, 2 min; temperature, 25 ± 0.3 °C. The data showed the mean in the measurement time.

#### 2.3.5. RB Solubility in Water

Various 0.05 mg/mL GM suspensions were prepared by dispersing the GM in ultrapure water with stirring for 10 min. After that, the suspension was sonicated for 1 min and then stirred for 10 min again. The suspension was centrifuged at 50,000 rpm (240,585× *g*) at 25 °C for 60 min (Himac CP80MX, Hitachi Koki Co., Ltd., Tokyo, Japan). The supernatant was analyzed by HPLC after filtration through a 0.2 μm filter. The HPLC system consisted of a pump (PU-22089), UV detector (UV-2075), and column oven (Co-2067) (JASCO Corporation, Tokyo, Japan). HPLC was performed under the following conditions: mobile phase, 0.1 M phosphate buffer solution and acetonitrile (3:1); λmax, 326 nm; flow rate, 1.0 mL/min; column temperature, 30 °C; C18 analytical column (4.6 mm × 150 mm; InertSustain® C18, GL Sciences Inc., Tokyo, Japan).

#### 2.3.6. Evaluation of Dispersion Stability in Water

Various 0.2% *w/v* GM suspensions were prepared by dispersing the GM in ultrapure water with stirring for 10 min. After that, the suspension was sonicated for 1 min and then stirred again for 10 min before each measurement. The dispersive stability in water was measured at room temperature (approximately 25 °C) using a Turbiscan MA 2000 (Formulaction, Toulouse, France). The analysis



time was 1 day, and the timing of the scan was once every hour. RB mouthwash was used as a reference formulation.

### 2.3.7. Evaluation of Mucus Permeation by RB Suspension

Various 0.2% *w/v* GM suspensions were prepared by dispersing the GM in ultrapure water with stirring for 10 min. After that, the suspension was sonicated for 1 min and then stirred for 10 min again. The mucus permeation of RB was measured at 25 °C using a 5.0 μm centrifugal tube filter (Ultrafree®-MC-SV, Merck Millipore, Burlington, USA). Then, 50 μL of 0.2 *w/v*% mucus solution was distributed on the poly vinylidene difluoride (PVDF) membrane by centrifugation at 500 rpm for 1 min (Sorvall Legend Micro 17R, Thermo Fisher Scientific, Waltham, MA, USA). Then, 300 μL of the suspension was added to the filtration chamber and centrifuged at 500 rpm for 45 min. The amount of RB that penetrated the mucin layer was reflected in the RB concentration of the filtrate. The permeation ratio (P) was calculated according to the following equation:

$$P = \text{(amount of RB with mucin layer)/(amount of RB without mucin layer)} \times 100 \tag{1}$$

### 2.3.8. Statistical Analysis

The data in Section 2.3.7 are expressed as mean ± standard deviation (SD). Statistical analysis was performed by one-way analysis of variance (ANOVA). Dunnett's test was used to assess differences from the control. A value of $p < 0.05$ was considered statistically significant.

## 3. Results and Discussion

The changes in the crystalline states of various samples were evaluated by powder X-ray diffraction (PXRD). The PXRD patterns of various samples are shown in Figure 1 and Supplementary Figures S1–S5. The RB crystals had characteristic peaks at $2\theta(°) = 12.3, 14.7, 17.9$, and 21.8 (Figure 1a) [47]. The characteristic peaks of RB are shown for each physical mixture (PM) (Figure 1b,c and Supplementary Figures S2a, S3a and S5a). However, the intensity of the peaks decreased upon grinding with each polymer (Figure 1b,c and Supplementary Figures S1, S2b–d, S3b–d, S4 and S5b–d). Furthermore, the intensity of the peaks decreased with the extension of grinding time. In particular, the peaks due to RB crystals disappeared in GM(1:3:1)-45 and GM(1:5:1)-45 when HPC-SSL was used as a water-soluble polymer. In other words, the analysis showed a halo pattern, which suggests that RB existed in the samples in an amorphous form. On the other hand, the intensity of the peaks due to RB crystals decreased after grinding for 30 min with PVP K30 (Figure 1c), which also suggests that RB was present in the samples in amorphous form in GM(1:3:1)-30 and GM(1:5-:1)-30 when PVP K30 was used as a water-soluble polymer because the peaks due to RB crystals disappeared.

The intermolecular interaction between RB and various polymers was evaluated by ATR-FTIR spectroscopy. RB has three carbonyl groups (Figure 2), and their characteristic absorption peaks are shown at 1720–1700 cm$^{-1}$, 1650–1550 cm$^{-1}$, and 1400 cm$^{-1}$ [47]. It was reported that the peak at 1720 cm$^{-1}$ disappears if RB forms as salt [47]. In this study, because the peak at 1720 cm$^{-1}$ did not disappear in each sample, it was suggested that the RB and the sodium of SDS did not form a salt (Figure 3a,c). One of the peaks, at around 1640 cm$^{-1}$ in the FT-IR spectra, was focused on the intermolecular interaction between RB and each polymer in this study. Figure 3a,c shows the FT-IR spectra of various samples. Furthermore, Figure 3b,d and Supplementary Figures S6–S10 show the FT-IR spectra of the carbonyl group (around 1640 cm$^{-1}$) in PMs and GMs that were prepared with various polymers. In this study, for each polymer, no peak shift was observed when they were ground for various time intervals (data not shown). Additionally, no peak shifts were observed in any of the PMs, suggesting that there was no intermolecular interaction between RB and polymer (Figure 3 and Supplementary Figures S1–S10). On the other hand, a peak shift due to the stretching vibration of the carbonyl group was observed in GMs with HPCs, which shifted to a higher frequency (Figure 3b and Supplementary Figures S6–S8). Nearly the same results were obtained when PVPs were used as

grinding media. No peak shift of the carbonyl group was observed in PMs with PVPs, while in GMs with PVPs, a peak shift due to the stretching vibration of the carbonyl group was observed (Figure 3b and Supplementary Figures S9 and S10). These results suggest there were intermolecular interactions between RB and HPCs or PVPs, and that they were excited as solid dispersions in the mixtures.

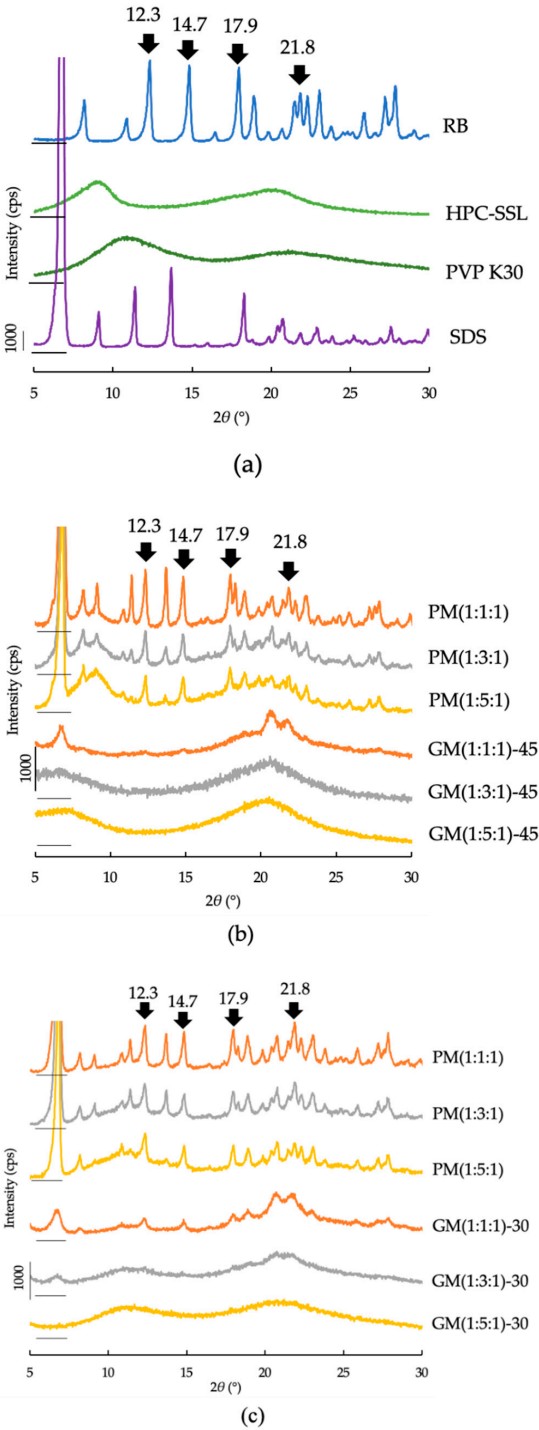

**Figure 1.** Powder X-ray diffraction (PXRD) patterns of samples. Black arrows with numbers show characteristic peaks of RB crystals. (**a**) RB crystals, hydroxypropyl cellulose (HPC)-SSL, polyvinylpyrrolidone (PVP) K30, and sodium dodecyl sulfate (SDS); (**b**) physical mixtures (PMs) and ground mixtures (GMs) with HPC-SSL in various mixing ratios (grinding time: 45 min); (**c**) PMs and GMs with PVP K30 in various mixing ratios (grinding time: 30 min).

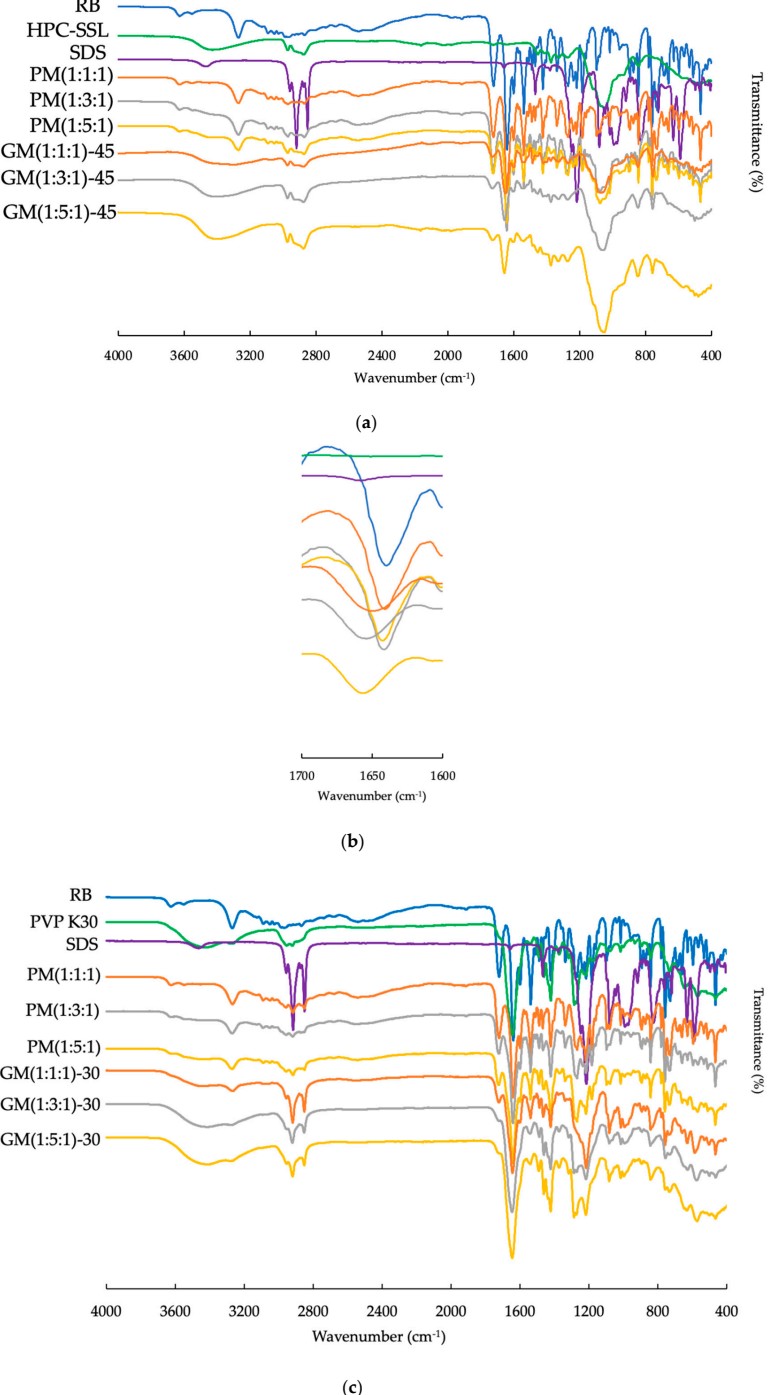

**Figure 2.** Chemical structural formula of RB. The carbonyl group (black arrow) was the focus of FT-IR.

**Figure 3.** *Cont.*

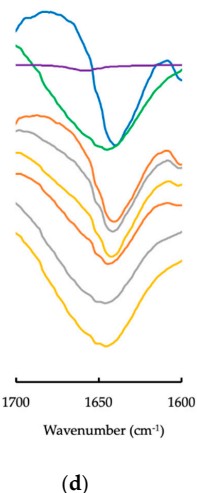

(**d**)

**Figure 3.** FT-IR spectra of samples. (**a**) PMs and GMs with HPC-SSL in various mixing ratios (grinding time: 45 min); (**b**) PMs and GMs with HPC-SSL in various mixing ratios (grinding time: 45 min) from 1600 to 1680 cm$^{-1}$; (**c**) PMs and GMs with PVP K30 in various mixing ratios (grinding time: 30 min); (**d**) PMs and GMs with PVP K30 in various mixing ratios (grinding time: 30 min) from 1600 to 1680 cm$^{-1}$.

The particle size, zeta potential, and viscosity of various samples suspended in water are shown in Figure 4, Tables 3 and 4.

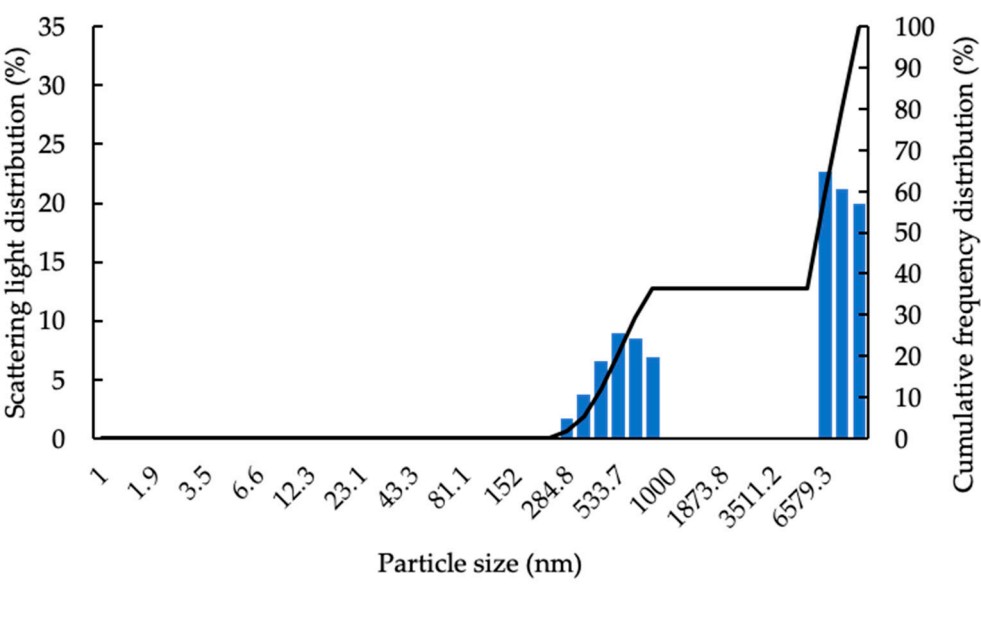

(**a**)

**Figure 4.** *Cont.*

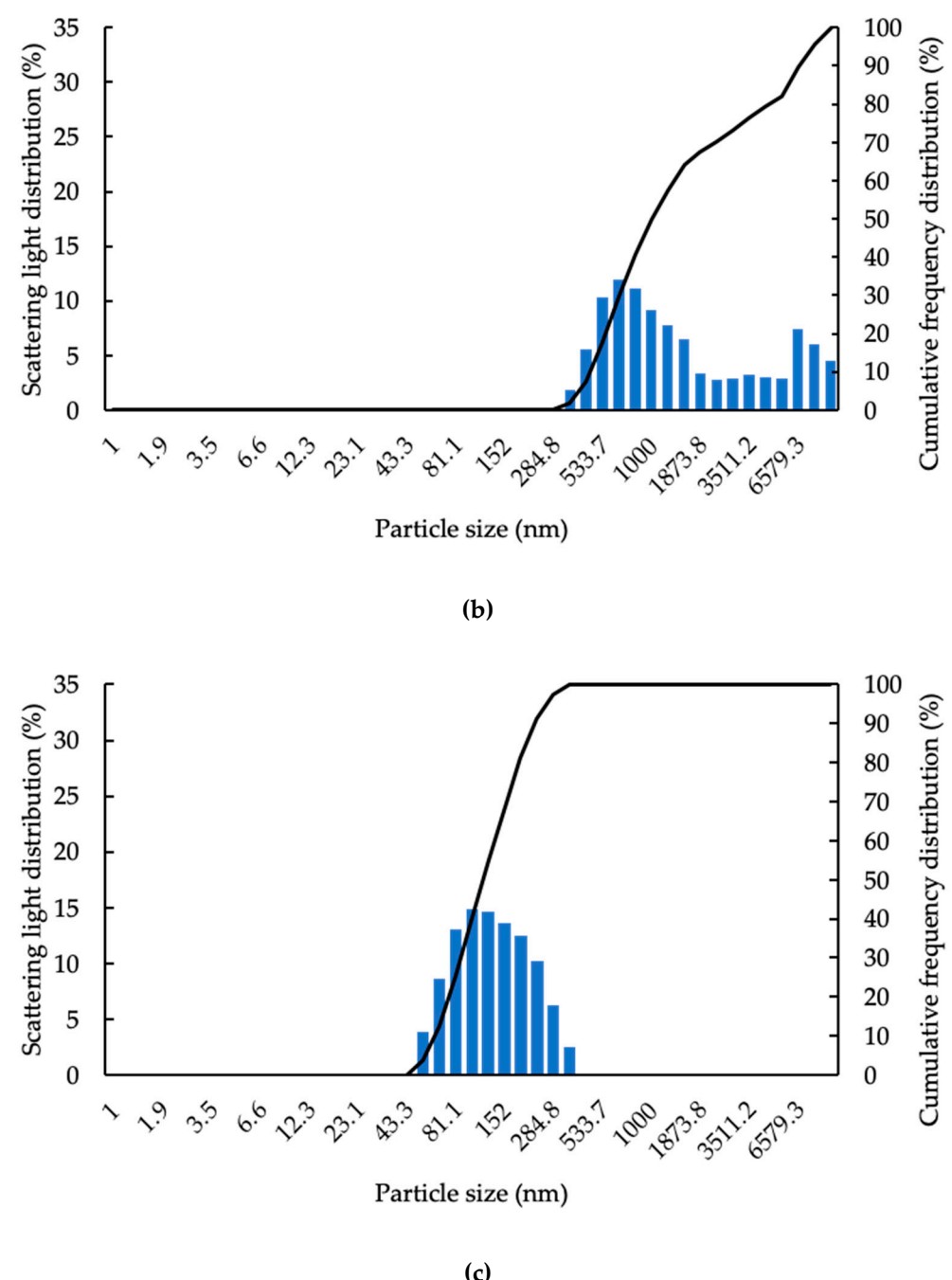

**(b)**

**(c)**

**Figure 4.** Particle size distribution of representative samples: (**a**) RB; (**b**) PM (1:5:1)-30 with PVP K30; (**c**) GM (1:5:1)-30 with PVP K30.

**Table 3.** Particle sizes of samples.

| Sample | Particle Size (nm) | | | | | | | | | |
|---|---|---|---|---|---|---|---|---|---|---|
| | HPC-SSL | | HPC-SL | | HPC-L | | PVP K30 | | PVP K90 | |
| | Mean | PDI | Mean | PDI | Mean | PDI | Mean | PDI | Mean | PDI |
| PM (1:1:1) | 1271.2 ± 194.2 | 0.485 ± 0.133 | 920.0 ± 238.9 | 0.339 ± 0.138 | 855.7 ± 101.7 | 0.612 ± 0.250 | 1520.3 ± 157.7 | 0.588 ± 0.133 | 1679.8 ± 614.5 | 0.600 ± 0.126 |
| PM (1:3:1) | 1679.7 ± 345.2 | 0.627 ± 0.193 | 904.6 ± 166.2 | 0.310 ± 0.105 | 996.2 ± 61.5 | 0.394 ± 0.110 | 1503.3 ± 225.4 | 0.641 ± 0.131 | 2016.5 ± 177.7 | 0.723 ± 0.194 |
| PM (1:5:1) | 796.5 ± 161.8 | 0.301 ± 0.064 | 925.2 ± 14.1 | 0.5132 ± 0.219 | 975.0 ± 79.4 | 0.539 ± 0.238 | 1869.0 ± 758.8 | 0.797 ± 0.450 | 1688.7 ± 778.6 | 0.593 ± 0.262 |
| GM (1:1:1)-15 | 243.9 ± 19.9 | 0.165 ± 0.005 | 318.2 ± 47.6 | 0.206 ± 0.045 | 364.5 ± 62.6 | 0.2378 ± 0.052 | 256.9 ± 36.1 | 0.169 ± 0.024 | 467.4 ± 99.1 | 0.296 ± 0.054 |
| GM (1:3:1)-15 | 220.0 ± 24.1 | 0.217 ± 0.030 | 212.3 ± 31.9 | 0.146 ± 0.025 | 264.6 ± 28.9 | 0.187 ± 0.017 | 217.7 ± 36.2 | 0.171 ± 0.016 | 1380.3 ± 658.5 | 0.587 ± 0.136 |
| GM (1:5:1)-15 | 229.3 ± 32.8 | 0.218 ± 0.035 | 209.7 ± 49.6 | 0.240 ± 0.079 | 274.8 ± 42.2 | 0.248 ± 0.055 | 186.4 ± 29.2 | 0.159 ± 0.013 | 1040.6. ± 171.6 | 0.591 ± 0.190 |
| GM (1:1:1)-30 | 221.4 ± 25.9 | 0.199 ± 0.060 | 239.4 ± 32.7 | 0.196 ± 0.071 | 309.2 ± 41.0 | 0.223 ± 0.038 | 222.9 ± 44.0 | 0.170 ± 0.016 | 378.3 ± 120.0 | 0.249 ± 0.040 |
| GM (1:3:1)-30 | 164.1 ± 16.0 | 0.226 ± 0.040 | 162.5 ± 4.2 | 0.168 ± 0.014 | 199.5 ± 3.7 | 0.202 ± 0.022 | 175.0 ± 32.5 | 0.177 ± 0.001 | 304.6 ± 56.5 | 0.271 ± 0.026 |
| GM (1:5:1)-30 | 166.8 ± 16.2 | 0.195 ± 0.072 | 171.4 ± 7.4 | 0.227 ± 0.025 | 175.2 ± 18.0 | 0.219 ± 0.031 | 145.0 ± 25.1 | 0.165 ± 0.016 | 451.6 ± 134.5 | 0.246 ± 0.036 |
| GM (1:1:1)-45 | 180.0 ± 49.1 | 0.217 ± 0.052 | 230.8 ± 46.0 | 0.246 ± 0.071 | 221.8 ± 34.6 | 0.206 ± 0.044 | 198.4 ± 50.07 | 0.181 ± 0.022 | 301.3 ± 68.6 | 0.284 ± 0.013 |
| GM (1:3:1)-45 | 112.4 ± 2.6 | 0.201 ± 0.033 | 130.5 ± 14.9 | 0.191 ± 0.071 | 173.5 ± 27.8 | 0.147 ± 0.011 | 140.0 ± 14.1 | 0.147 ± 0.011 | 509.9 ± 123.7 | 0.278 ± 0.016 |
| GM (1:5:1)-45 | 135.1 ± 40.8 | 0.206 ± 0.030 | 167.4 ± 11.5 | 0.191 ± 0.043 | 176.8 ± 13.0 | 0.232 ± 0.014 | 137.0 ± 22.6 | 0.128 ± 0.014 | 432.3 ± 8.0 | 0.236 ± 0.012 |

Data represent mean ± SD of three experiments.

**Table 4.** Zeta potential, viscosity, and solubility in water of samples.

| Sample | Zeta Potential (mV) | | Viscosity (mP•s) | | Solubility (µg/mL) | |
|---|---|---|---|---|---|---|
| | HPC-SSL | PVP K30 | HPC-SSL | PVP K30 | HPC-SSL | PVP K30 |
| PM (1:1:1) | −8.85 | −9.92 | 1.02 | 0.92 | 7.57 | 7.89 |
| PM (1:3:1) | −7.72 | −8.28 | 1.42 | 1.13 | 7.47 | 6.83 |
| PM (1:5:1) | −6.27 | −9.56 | 1.96 | 1.20 | 8.56 | 5.73 |
| GM (1:1:1)-15 | −9.19 | −13.93 | 1.01 | 1.03 | 11.40 | 12.33 |
| GM (1:3:1)-15 | −7.34 | −10.42 | 1.28 | 1.10 | 14.04 | 12.57 |
| GM (1:5:1)-15 | −7.18 | −8.41 | 1.78 | 1.16 | 16.89 | 15.09 |
| GM (1:1:1)-30 | −10.00 | −16.01 | 1.02 | 1.04 | 13.24 | 12.42 |
| GM (1:3:1)-30 | −6.60 | −11.78 | 1.20 | 1.08 | 17.64 | 14.04 |
| GM (1:5:1)-30 | −7.11 | −10.86 | 1.58 | 1.13 | 19.91 | 18.74 |
| GM (1:1:1)-45 | −8.82 | −13.51 | 0.99 | 1.00 | 14.78 | 17.07 |
| GM (1:3:1)-45 | −12.46 | −13.09 | 1.18 | 1.30 | 18.83 | 17.73 |
| GM (1:5:1)-45 | −11.02 | −11.656 | 1.48 | 1.20 | 21.09 | 24.66 |

The mean particle sizes of RB crystals and RB mouthwash were 1496.7 nm and 1978 nm, respectively (data not shown). However, the mean particle size decreased upon grinding with various polymers and SDS. Furthermore, the mean particle size of GMs with HPC-SSL or PVP K30 was 112.4 nm and 137.0 nm, respectively, the smallest for each polymer. In GMs with HPCs, the mean particle size decreased with the increasing mixing ratio of HPCs and extension of grinding time. Furthermore, GM with PVP K30 showed a decrease in particle size with the increasing mixing ratio of PVP K30 and extension of grinding time. In each polymer, the particle size depended on the molecular weight. In other words, when lower-molecular-weight HPCs or PVPs were used for grinding media, the particle size was smaller. As mentioned in the results of FT-IR, when RB and HPCs or PVPs were ground together, a solid dispersion formed, and monomolecular dispersed RB seemed to interact with the surrounding polymer chains of HPCs or PVPs. It was observed that when a mixture with the same weight ratio of the polymer was suspended in water, the particle size depended on the length of the polymer chain; that is, its molecular weight. Shudo et al. demonstrated that nanoparticles consisting of probucol, PVP, and SDS formed micelle-like complexes covered as layered structures [48]. It was considered that nanoparticles consisting of HPC or PVP were covered with molecular interactions. In this study, it was considered that results showed a decreased particle size if the molecular weight was low.

The zeta potential values of samples using HPCs or PVPs are shown in Table 4. In PMs, the zeta potential of each sample shifted more toward the positive side than that of RB crystals. This suggests that PM characteristics such as solid dispersion, despite the intermolecular interaction, were not observed in the FT-IR spectra. The zeta potential of each sample changed very little when the sample was ground. This suggested that the surface of the nanoparticles was covered with polymers by the molecular interaction between RB and polymers. In addition, it suggested that SDS adsorbed onto the drug nanoparticle surface without molecular interaction in the suspension. However, the structure of nanoparticles should be clarified by a measurement method, such as NMR, in the future. As mentioned above, the solution needs to have a zeta potential less than −10 mV in order for there to be adherence between particles and oral mucosa for the repulsion of electrostatic force [26]. The suspension of RB crystals showed a zeta potential of −26.1 mV (data not shown). The zeta potential when samples were ground shifted a little to the positive side, e.g., −9.2 and −11.6 (Table 4). It was considered that these changes appeared because RB and polymers interacted when they were ground together. Furthermore, the suspension was considered to permeate the mucous layer because GMs can adhere to the oral mucosa, since almost all GM suspensions had a zeta potential of more than −10 mV.

In this study, we used only 0.2% *w/v* of each sample solution in the viscosity measurements, so that no samples were viscous and were almost equal to water, which was 0.99 mPa·s at 25 °C (Table 4). From the results, it was considered that the viscosity of each sample solution did not affect the stability of the dispersion.

We considered that when RB suspension is applied as a mouthwash for stomatitis, improved RB solubility plays an important role in improving its dispersibility. The RB solubility of various samples suspended in water is shown in Table 4. The solubility of GMs increased about 1.5- to 4.3-fold compared to RB crystals and PMs. There were no differences in RB solubility between HPCs and PVPs or molecular weights. Furthermore, solubility improved with increased polymer concentration. In particular, solubility was higher when the grinding time was longer for each polymer. As these results show, water-soluble polymers and surfactants contributed to supersaturation to suppress crystallization in this case [49]. Furthermore, extending the grinding time improved the solubility of RB because the solubility of GMs improved with increased grinding time. It was thought that the particle size of RB crystals was decreased because of the increased amount of impact of the ball in the mill, and the regular array structure of the crystals was broken by grinding.

We next focused on dispersibility in water and mucus permeation of RB suspension. We evaluated GM (1:3:1)-45 with HPC-SSL and GM (1:5:1)-30 with PVP K30, which achieved almost minimum particle sizes in all samples. The RB mouthwash was used as a reference. Figure 5 shows the hourly change of transmission and backscattering of the samples over 24 h. In the figure, the height from the

bottom to the upper side (from left to right) of the sample tube is shown on the *x*-axis, and the change in transmission or backscattering of the samples is shown on the *y*-axis. A Turbiscan MA 2000 consists of a pulsed near-infrared light source and two synchronous detectors with heads that move up and down along the cylindrical tube, and the detection angles of transmission and backscattered light are 180° and 45° [50]. In the RB mouthwash, an increase in transmission was shown early, which means that it was not dispersed enough. On the other hand, there was almost no change in transmission and backscattering after 24 h. Furthermore, there were more such changes in PVP K30 than in HPC-SSL. However, the changes were small, and it was clear that these suspensions had good dispersibility after 24 h.

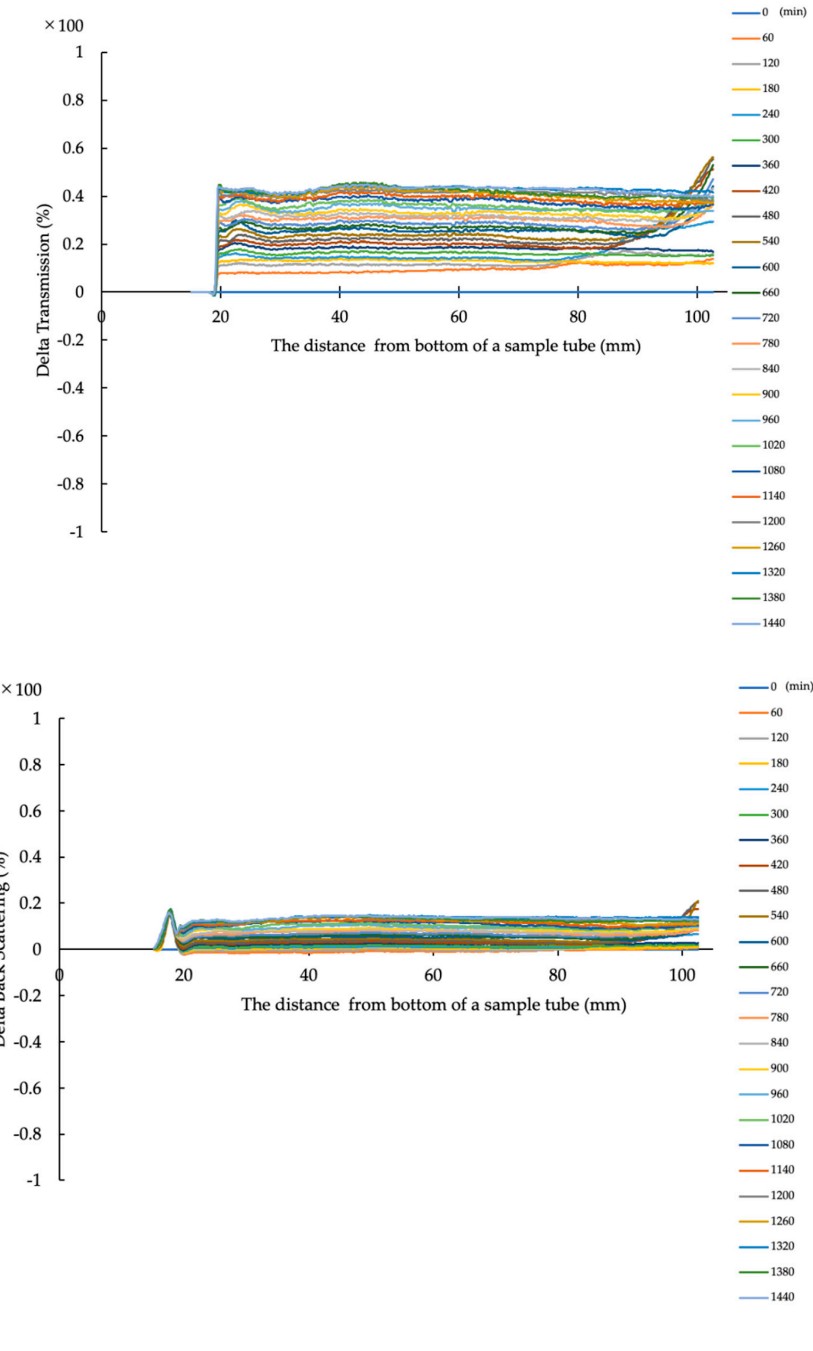

(a)

**Figure 5.** *Cont.*

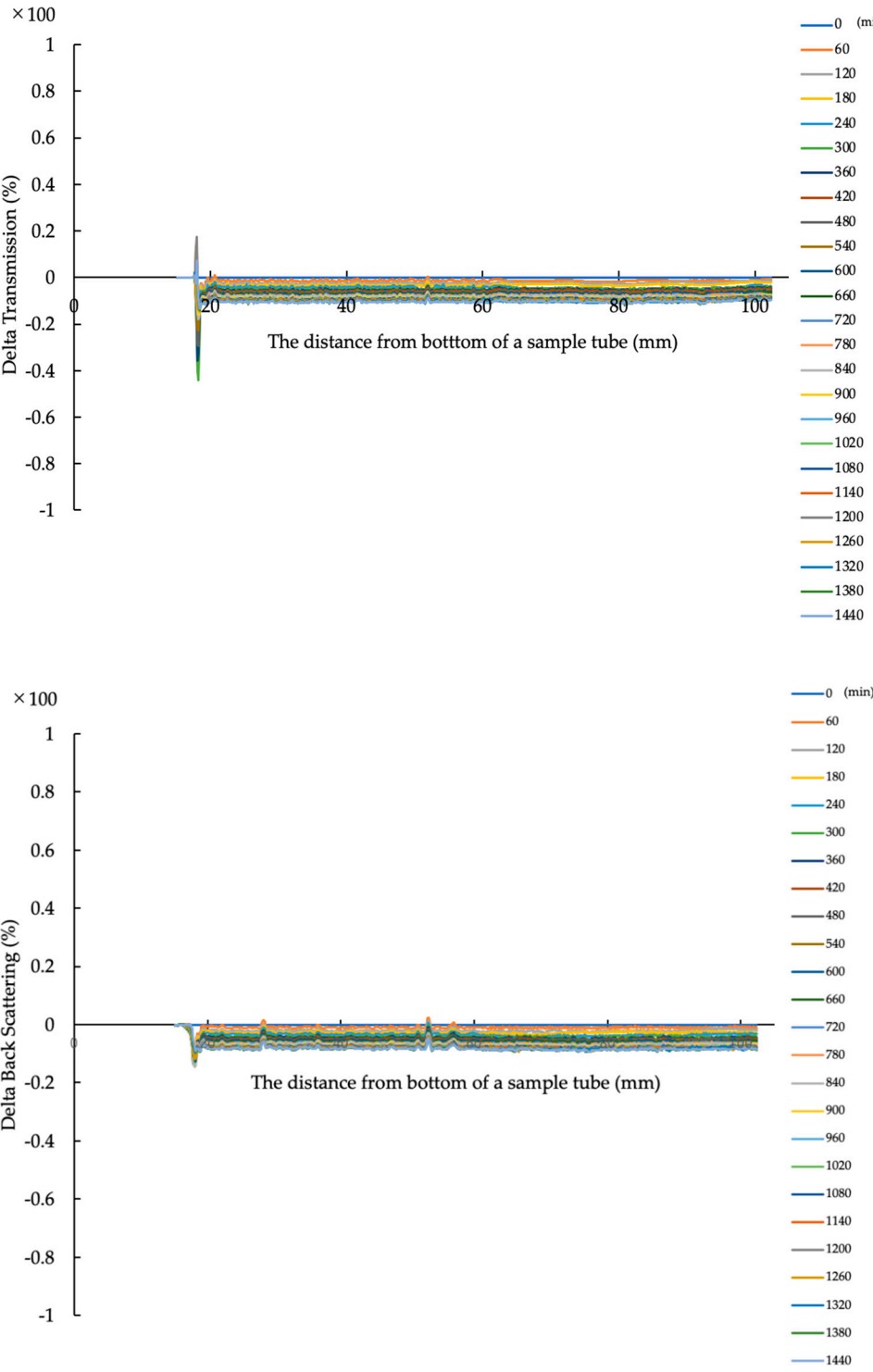

**(b)**

**Figure 5.** *Cont.*

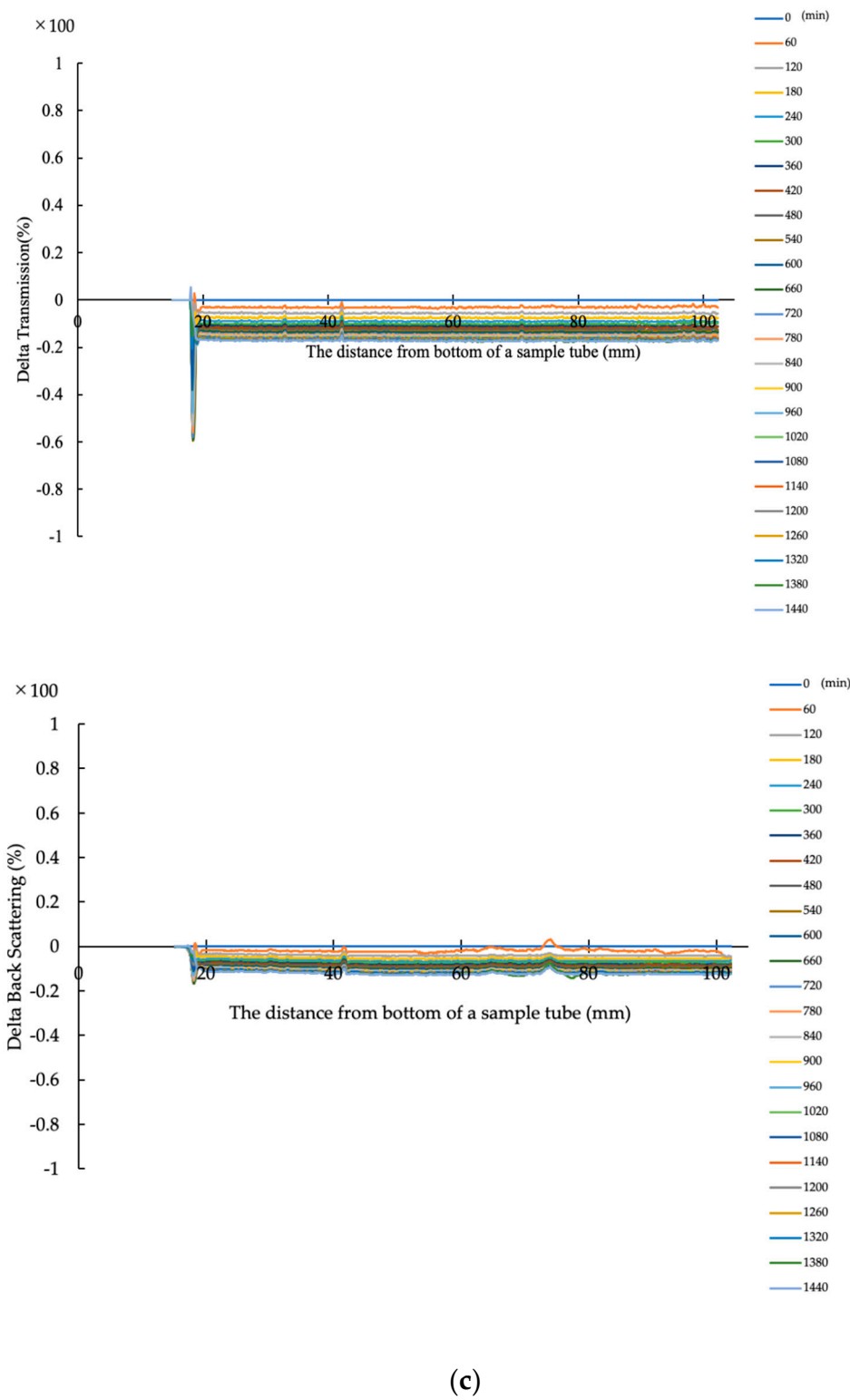

(c)

**Figure 5.** Changes of transmission and backscattering patterns of samples: (**a**) RB mouthwash, (**b**) GM (1:3:1)-45 with HPC-SSL, and (**c**) GM (1:5:1)-30 with PVP K30.

Liu et al. reported that RB could not reach the oral mucosa layer because there is mucus consisting of a mucin layer [51,52]. In this study, we developed a new evaluation method for mucus permeation on mucin layers. Traditionally, mucus permeation has been evaluated using Transwell or side-by-side diffusion cells [53]. In each method, the donor and receptor compartments must be full of solution.

Then, the pH values of the donor and receptor solutions equalize over time if the pH of the solution is 7.0, equal to that of mucus. However, RB can dissolve in a pH 7.0 solution despite the poor solubility of the drug. Therefore, RB may flow from the receptor to the donor compartment in the Transwell or side-by-side diffusion cell method, making the evaluation of mucus permeation impossible. In this study, we focused on a centrifugal filtration device in which the mucin solution fills only the donor compartment (Scheme 1). Figure 6 shows the mucus permeation of RB mouthwash, GM (1:3:1)-45 with HPLC-SSL, and GM (1:5:1)-30 with PVP K30 suspension, which were prepared at 0.2% *w/v* of RB concentration. The permeation of RB mouthwash and GM suspensions was approximately 36% and 90%, respectively. These results show that GM suspensions could improve the healing or protective effect against stomatitis because the RB particles can permeate the mucin layer easily.

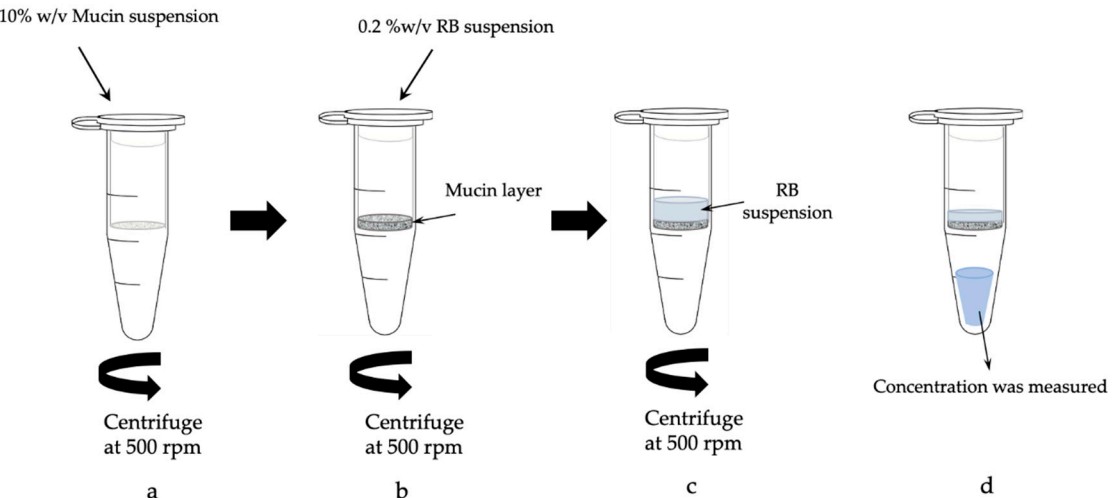

**Scheme 1.** Method of mucus permeation using centrifugal filtration device with mucin solution.

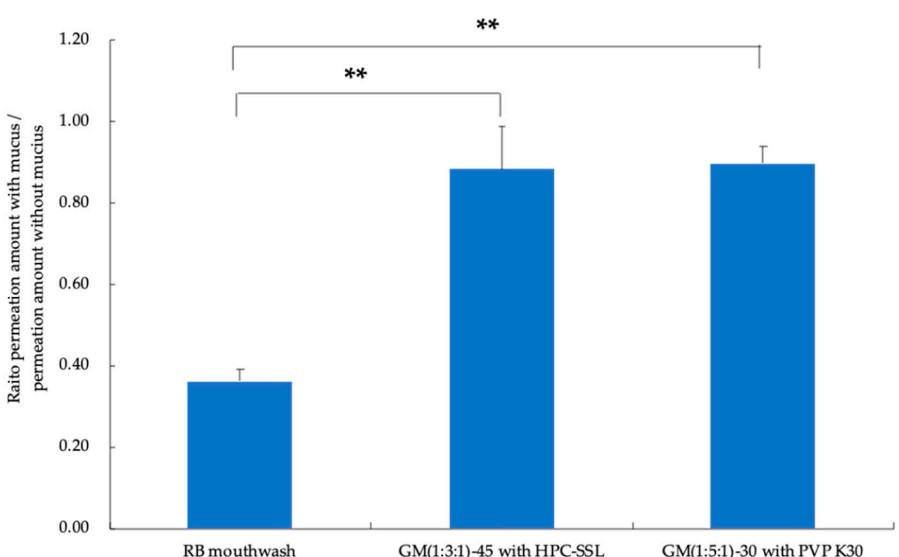

Values are shown as mean ± SD (*n* = 3). ** *p* < 0.01 vs. RB

**Figure 6.** Mucus permeation evaluation of RB mouthwash, GM (1:3:1)-45 with HPC-SSL, and GM (1:5:1)-30 with PVP K30.

## 4. Conclusions

In this study, we focused on RB nanoparticles using a mixer ball mill to improve the dispersibility and permeation of the oral mucus layer. The mean particle size of RB in the suspension of around

100 nm was the minimum when RB crystals were ground with various weight ratios of PVP or HPC and SDS. The PXRD analysis of these samples showed a halo pattern. In other words, the diffraction peaks of RB crystals disappeared, suggesting the existence of an amorphous state of RB. Furthermore, peak shifts of the carbonyl group, which showed stretching vibration, were observed, suggesting intermolecular interactions between RB and PVP or HPC. In addition, it was shown that RB nanoparticles in suspension had good dispersibility and high permeation of the oral mucus layer.

From these results, we concluded that RB nanoparticles ground with PVP or HPC and SDS, prepared using a desktop-type mixer ball mill, can be applied as a formulation for the treatment and/or prevention of stomatitis. In this study we used SDS as a surfactant, which irritates the mucous membrane; therefore, we used a small amount, which should not produce this effect. However, the use of other surfactants that do not cause irritation to the mucous membrane should also be considered in future studies. Furthermore, we used a stainless-steel ball and jar for grinding. However, there were potential concerns regarding the contamination of the stainless-steel for wear. Therefore, for a future study, we should check that using other methods, such as atomic spectroscopy, inductively coupled plasma mass spectrometry, and so on.

**Supplementary Materials:** The following are available online at http://www.mdpi.com/2504-5377/4/4/43/s1, **Figure S1.** PXRD patterns of samples using HPC-SSL as a water-soluble polymer: GMs of various mixing ratios with grinding time of (**a**) 15 min and (**b**) 30 min. **Figure S2.** PXRD patterns of various samples using HPC-SL as a water-soluble polymer. (**a**) RB crystals, HPC-SL, SDS, and PMs. GMs of various mixing ratios with grinding time of (**b**) 15 min, (**c**) 30 min, and (**d**) 45 min. **Figure S3.** PXRD patterns of samples using HPC-L as water-soluble polymer. (**a**) RB crystals, HPC-L, SDS, and PMs. GMs of various mixing ratios with grinding time of (**b**) 15 min, (**c**) 30 min, and (**d**) 45 min. **Figure S4.** PXRD patterns of samples using PVP K30 as a water-soluble polymer. GMs of various mixing ratios with grinding time of (**a**) 15 min and (**b**) 45 min. **Figure S5.** PXRD patterns of samples using PVP K90 as water-soluble polymer. (**a**) RB crystals, HPC-L, SDS, and PMs. GMs of various mixing ratios with grinding time of (**b**) 15 min, (**c**) 30 min, and (**d**) 45 min. **Figure S6.** FT-IR spectra of samples using HPC-SSL as a water-soluble polymer. GMs of various mixing ratios with grinding time of (**a**) 15 min and (**b**) 30 min. **Figure S7.** FT-IR spectra of samples using HPC-SL as a water-soluble polymer. (**a**) RB crystals and PMs. GMs of various mixing ratios with grinding time of (**b**) 15 min, (**c**) 30 min, and (**d**) 45 min. **Figure S8.** FT-IR spectra of samples using HPC-L as a water-soluble polymer. (**a**) RB crystals and PMs. GMs of various mixing ratios with grinding time of (**b**) 15 min, (**c**) 30 min, and (**d**) 45 min. **Figure S9.** FT-IR spectra of samples using PVP K30 as a water-soluble polymer. GMs of various mixing ratios with grinding time of (**a**) 15 min and (**b**) 45 min. **Figure S10.** FT-IR spectra of samples using PVP K90 as a water-soluble polymer. (**a**) RB crystals and PMs. GMs of various mixing ratios with grinding time of (**b**) 15 min, (**c**) 30 min, and (**d**) 45 min. **Table S1.** Zeta potential, viscosity, and solubility in water of samples using HPC-SL, HPC-L, and PVP K90 as water-soluble polymers.

**Author Contributions:** Conceptualization, Y.K. and T.H.; methodology, Y.K. and T.H.; validation, Y.U. and F.Y.; formal analysis, Y.K.; investigation, Y.U., F.Y., Y.K., and N.I.; data curation, Y.U., F.Y., Y.K., and N.I.; writing—original draft preparation, Y.K. and Y.U.; writing—review and editing, T.H.; supervision, T.H.; project administration, T.H. All authors have read and agreed to the published version of the manuscript.

**Funding:** This research received no external funding.

**Acknowledgments:** We would like to thank Nippon Soda Co., Ltd. for providing HPC-L, HPC-SL, and HPC-SSL. Additionally, we would like to thank Editage (www.editage.com) for English language editing.

**Conflicts of Interest:** The authors declare no conflict of interest.

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
