# Peer review of "Preparation and Evaluation of Rebamipide Colloidal Nanoparticles Obtained by Cogrinding in Ternary Ground Mixtures"

_colloids, doi:10.3390/colloids4040043_

Round 1
Reviewer 1 Report
Please see attached.

Author Response
#Reviewer 1
Thank you very much for reviewer’s raising important issues and providing us helpful suggestions. We have performed additional experiments and extensively revised our manuscript in the light of reviewer’s comments.
- This manuscript needs to be thoroughly proofread, preferably by a native English speaker. There are some sentences and paragraph that do not make sense or add a lot to the manuscript e.g. paragraph on lines 66-68. I know what you have tried doing here but this needs a little more information/description.
>Response
We appreciate your meaningful suggestion. According to the reviewer’s comment, we added a little more description about approaches for preparation of nanoparticles.
- Line 98- refrigerated should be frozen
>Response
Thank you for your suggestion. We changed the word.
- Materials and method section: In general, details in methods is either missing or insufficient.
Please add as much detail in every section as possible. Some of the examples are as follows:
- PXRD - 2 theta range
- ATR - method is not descriptive enough, rewrite and provide all required information clearly.
- Particle Size number of measurements and repeats etc.
- Viscosity - how did you run these measurements e.g. against temp, time, shear rate etc?
again any repeats? lacks general details and not very well written.
- Mucus penetration concentration of mucus solution?
>Response
Thank you for your suggestion. We added more information about each measurement.
- Result and discussion: again, no real discussion I am afraid:
- There is nothing wrong with the interpenetration of XRD and FTIR data but no references used to justify drawn conclusions.
> Response
Thank you for your suggestion. We interpreted our results by referring to findings and understanding described in various references, then we added the description in PXRD section and FTIR section.
- Table 3 - No real discussion on why do you see this particular trend? Why SSL appears to be the best choice when compared to SL and L or PVPs?
Also why do you obtain smaller particles with K30 in comparison to K90.
The results appear to be pretty obvious and reflect the effect of molecular weight of the polymers used but this needs to be clearly discussed. Use relevant references
With respect to numbers; one can never be as accurate as 1176.5 nm, please round these numbers and specify associated error (e.g. plus/minus SD)
>Response
Thank you for your suggestion. We improved the discussion for these results. And, we showed the numbers as mean ± SD in Table 3.
- Table 4 - same applies to the data presented to the 2 decimal points here, surface charge data using a zeta-sizer is at best an estimate so presenting these numbers to 2 dp does not make any sense.
Again, I am assuming these are mean of few measurements, please specify associated error.
Also be careful with the second dp for solubility unless you have clearly performed these experiments with such sensitivity.
>Response
Thank you for your suggestion. We revised the decimal points at zeta-sizer.
- Rewrite the text in lines 211-216
> Response
Thank you for your suggestion. We rewrote this paragraph and described based on the details of results inline 905-919.
The zeta potentials of samples using HPCs or PVPs are shown in Table 4. In PMs, the zeta potential of each sample showed to shift more positive side than that of RB crystal’s. This suggests the PMs are statuses like solid dispersion despite the intermolecular interaction were not observed in the FT-IR spectra. The change in zeta potential of each sample was very little when the sample was ground. The surface of the nanoparticles suggested to be covered with polymers by molecular interaction between RB and polymers. In addition, it was suggested SDS adsorb onto the drug nanoparticles surface without molecular interaction in the suspension. However, the structure of nanoparticles should be made clear by some measurement, such as NMR in the future. As mentioned above, it needs to be less than -10 mV of the zeta potential to the solution in order to adhere between particles and oral mucosa for the repulsion of an electrostatic force [26]. Suspension of RB crystals was shown -26.1 mV on the zeta potentials (data not shown). The zeta potentials of when samples were ground were shifted to a little positive side, e.g., -9.2, and -11.6, so on (Table 4). It was considered that these changing appeared because RB and polymers interacted when these were ground together. On the other hand, it was considered to permeate the mucous layer because GMs can adhere to the oral mucosa as almost GM suspensions were more than -10 mV on these zeta potentials.
- Lines 217-218 ok but why? Is it expected/unexpected? Is it a surprise considering these suspensions are prepared in water or should you expect some differences in the viscosity due to polymers used?
> Response
Thank you for your suggestion. In this study, we prepared samples at low concentration of polymer solutions. Therefore, we rewrote this as follows in line 920-923.
In this study, we used only 0.2% w/v of each sample solution in the viscosity measurements, then none of the samples were viscous and they were almost equal to water, which is 0.99 mPa・s at 25°C (Table 4). From the results, it was considered that viscosity of each sample solution did not affect to stability of dispersion.
- Line 220 - What is RB's solubility in water?
Discuss the data in detail: include text on which polymer resulted in highest and
lowest solubility and explain why.
> Response
Thank you for your suggestion. We rewrote and added the interpretation this paragraph.
We considered that when RB suspension is applied as mouthwash for stomatitis, improvement of RB solubility plays an important role to improve the dispersibility of RB. The RB solubility of various samples suspended in water are shown in Table 4. The solubility of GMs increased about 1.5- to 4.3-fold compared to that of RB crystals and PMs. There were no difference in solubility of RB between HPCs and PVPs or each molecular weight. Especially, the solubility improved with increasing in each polymer concentration. Among them, the solubility was shown higher when grinding time was long in each polymer. As these results show, water-soluble polymers and surfactants contribute to supersaturation for suppressing crystallization in this case [49]. Furthermore, extending the grinding time improved the solubility of RB because the solubility of GMs improved with increasing grinding time. It was thought that the particle size of RB crystals was decreased because the number of impacts of the ball in the mill increased, and the regular array structure of RB crystals was broken by grinding.
- Figure 4 - Can these figures be any clearer? Any possibility of plotting these yourself on Excel or origin for further clarity?
> Response
Thank you for your suggestion. We changed the figures which are clearer.
- I really liked the mucus permeation setup but could centrifugation at 500 rpm is just too harsh? How about using a shaker instead, wouldn't that represent oral cavity more closely to centrifugation?
> Response
Thank you very much for your opinion. We tried to some centrifugation rates then we checked the surface to take images using SEM. We were able to see the mucus layer on the filter in this speed. From these reasons, we selected this speed in this study.

Reviewer 2 Report
The presented paper is partially suitable for the Journal aims and scopes. The major part of the manuscript is dedicated to the structural studies of RB. Besides the overall quality of the paper is poor due to the presentation and description of the results. A major revision is needed and additional analysis is recommended especially SEM imaging of the samples. Detailed comments are listed below:
- 21 what does it mean 'halo patterns'?
- 29 change for the "one of the most frequent"
- 30 could you check recent reports (i.e. 2019)
- 54 only in Japan and the whole of Japan? Or it is the author's idea to prepare it. Can some other mouth liquids which containing RB are available on the market?
- 54 - 65 this section is more suitable for results and discussion. If the tablets are commercially available I don t feel the problem? Could you explain that deeper? Maybe preparation conditions were wrong?
- 66 and 69 make one paragraph.
Why did you use SDS for possible medical applications?
Materials and methods:
Please identify the main compound in Mucosta®ï¸Ž Tablets Simple syrup (commercial name) what kind of Preservative?
- 93 what kind of glass vial was used and what was their volume. Why the vortex mixer was used?
Please change the description presented in table 1 it is difficult to read how the samples were prepared.
l.105 how the samples were prepared for FTIR measurements? Did you use ATR?
- 110 what was the detection angle of scattered light and wavelength of the light source?
Powder X-ray 152 diffraction (PXRD)
--> please identify the peaks on the graphs
--> the graphs should be enlarged
The crystalline size determined by XRD should be presented - then you can make the discussion about the size.
Please change the name of HPLC - it usually dedicated to chromatography.
What do the Japan signs mean on the paragraphs when you mark the curves by cursor?
FTIR
- please state how the baseline was made.
- please give the full spectra not only in the selected range.
- add spectra of each "pure" component
Particle size determination
There is a lot of problems with the DLS measurement of solid particles. Besides, I like this technique as a rapid and general method for particle size determinations. Please give detailed descriptions of how the nanoparticle suspensions were prepared.
Some general studies and tips and tricks which can improve your studies you can find here:
Nanoparticles Size Determination by Dynamic Light Scattering in Real (Non-standard) Conditions Regulators - Design, Tests, and Applications https://link.springer.com/chapter/10.1007/978-3-030-39867-5_13
Microstructure and Electrochemical Properties of Refractory Nanocrystalline Tantalum-based Alloys - especially here you can find an alternative method for the DLS measurement interpretation.
Please add the graphs of DLS measurements and SD.
Fig. 4 should be totally changed because the quality is poor.
Please give some remarks about possible contaminations from metal balls during the ball milling process. Did you check this i.e. using atomic spectroscopy?
Author Response
Thank you very much for reviewer’s raising important issues and providing us helpful suggestions. We have performed additional experiments and extensively revised our manuscript in the light of reviewer’s comments.
The presented paper is partially suitable for the Journal aims and scopes. The major part of the manuscript is dedicated to the structural studies of RB. Besides the overall quality of the paper is poor due to the presentation and description of the results. A major revision is needed and additional analysis is recommended especially SEM imaging of the samples. Detailed comments are listed below:
- 21 what does it mean 'halo patterns'?
- 29 change for the "one of the most frequent"
- 30 could you check recent reports (i.e. 2019)
- 54 only in Japan and the whole of Japan? Or it is the author's idea to prepare it. Can some other mouth liquids which containing RB are available on the market?
- 54 - 65 this section is more suitable for results and discussion. If the tablets are commercially available I don t feel the problem? Could you explain that deeper? Maybe preparation conditions were wrong?
- 66 and 69 make one paragraph.
> Response
Thank you for your suggestions. We revised regarding 1 to 6 along your comments.
Especially, paragraph of line 54-65 were completely rewrote as follows (line 59-65 in revised manuscript):
RB has been reported to have effective to stomatitis caused by the chemotherapy or radiotherapy. Actually, the RB has been provided as one of gastric mucosal protectants in Japan. However, mouth washes containing RB (RB mouthwash) have never been as commercial products, thus RB mouthwash for curing stomatitis have been prepared as in-hospital formulations by pharmacists at many hospitals in Japan. In these cases, to prepare the mouth wash, RB tablets are ground suspended in water. The preparation process is not difficult, however, when once it results in precipitation, it is not easy for patients to re-disperse.
Why did you use SDS for possible medical applications?
> Response
Thank you for your suggestion. We preferred to use a solid surfactant because we considered preparation of nanoparticles using a mixer ball mill. As mentioned in Conclusion, it irritates the mucous membrane. However, we have used a small amount that should not produce this effect in this study.
Materials and methods:
Please identify the main compound in Mucosta®ï¸Ž Tablets Simple syrup (commercial name) what kind of Preservative?
> Response
I’m sorry it’s confused. Mucosta Tablets Simple syrup isn’t commercial products. However, we added preservative in the Table 1.
l.93 what kind of glass vial was used and what was their volume. Why the vortex mixer was used?
> Response
Thank you for your suggestions. We added about glass vials. And we used VORTEX mixer when the PMs were prepared because the mixer doesn’t give physically stimulate to the samples.
Please change the description presented in table 1 it is difficult to read how the samples were prepared.
> Response
Thank you for your suggestions. We added as sentences how the samples were prepared in line 132-143 in revised manuscript as follows:
“RB mouth wash” was prepared by the Department of Pharmacy in Kashiwa City Hospital (Kashiwa, Japan) as the reference formulation, and the prescription is shown in Table 1. Six tablets of Mucosta® Tablets including 100 mg as RB were ground by a tablet grinder (Iwatani Corporation, Tokyo, Japan) for 30 s. After grinding, the powder sieved through a No. 60 mesh sieve. The obtained powder was weight, then Citric acid (Kozakai Pharmaceutical Co., Ltd., Tokyo, Japan), L-glutamine (Nipro Corporation, Osaka, Japan), and simple syrup were added to that as additives. Additionally, dextrin (Toromeiku clear, Meiji Co., Ltd., Tokyo, Japan) as a thickening agent, 2 mL of preservation solution containig propyl or methyl p-hydroxybenzoate (Kanto Chemical Co., Inc. Tokyo, Japan) was added to the formulation. Furthermore, distill water was added to be a total 300 mL of the formulation.
l.105 how the samples were prepared for FTIR measurements? Did you use ATR?
> Response
Thank you for your suggestions. Yes, we used ATR. Then we added it in the method.
l.110 what was the detection angle of scattered light and wavelength of the light source?
> Response
Thank you for your suggestions. We added these in sentences in the method.
However, as for the angle of light scattering, since we could not find from the instruction manuals, we have been asking the company. If this manuscript would be accepted, we would like to add the angle to the proof reprints.
Powder X-ray 152 diffraction (PXRD)
--> please identify the peaks on the graphs
--> the graphs should be enlarged
> Response
Thank you for your suggestions. We polished and identified the peak in Figure of PXRD.
The crystalline size determined by XRD should be presented - then you can make the discussion about the size.
> Response
Thank you for your suggestions. I’m sorry we didn’t discuss about the crystalline size. So, we’ll consider that in future study.
Please change the name of HPLC - it usually dedicated to chromatography.
> Response
Thank you for your suggestions. It was miss typed. We rewrote it.
What do the Japan signs mean on the paragraphs when you mark the curves by cursor?
FTIR
- please state how the baseline was made.
- please give the full spectra not only in the selected range.
- add spectra of each "pure" component
> Response
Thank you for your suggestions. We improved the figures.
Particle size determination
There is a lot of problems with the DLS measurement of solid particles. Besides, I like this technique as a rapid and general method for particle size determinations. Please give detailed descriptions of how the nanoparticle suspensions were prepared.
Some general studies and tips and tricks which can improve your studies you can find here:
Nanoparticles Size Determination by Dynamic Light Scattering in Real (Non-standard) Conditions Regulators - Design, Tests, and Applications https://link.springer.com/chapter/10.1007/978-3-030-39867-5_13
Microstructure and Electrochemical Properties of Refractory Nanocrystalline Tantalum-based Alloys - especially here you can find an alternative method for the DLS measurement interpretation.
> Response
Thank you for your suggestions. We described the details about the preparation of nanoparticles. As in future study, we would like to get the full form about the nanoparticle like mentioned above, we would like to interpret the results by referring it.
Please add the graphs of DLS measurements and SD.
> Response
Thank you for your suggestions. We performed additional experiment and the obtained results were listed in Table 3.
Fig. 4 should be totally changed because the quality is poor.
> Response
Thank you for your suggestions. We improved the graphs.
Please give some remarks about possible contaminations from metal balls during the ball milling process. Did you check this i.e. using atomic spectroscopy?
> Response
Thank you for your suggestions. Throughout in this study we could not find any flakes from metal balls in SEM, however, we didn’t check it by atomic spectroscopy. So, we would like to confirm that in the future study.

Reviewer 3 Report
The manuscript about the nanoparticulation of rebamipide contains useful scientific information. In the current form, it requires some improvements before the appearance.
- Although the "Abstract" resolves the most used abbreviations, which is useful for those who are interested in abstract reading only, it is necessary to do also the same in the text body.
- In the keywords, 4 of 6 are the repetition from the title. This is unacceptable.
- Although the structure of rebamipide is shown in Figure 2, its earlier appearance, e.g., in the "Introduction" would be more advantageous.
- The first appearance of HPC-SSL is in line 81. The referee accepts the supplier's nickname, which appears in many publications, but it would be nice if a scientist one time gets the afford to resolve the meaning of "SSL".
- Although the description of the milling machine with the name of the manufacturer is correct, it would be useful to mention that this is a mixing mill, for easier reading. It is not expectable for the reader to surf on the internet to find what the equipment nickname means.
- In section 2.3.2, the IR spectrometer resolution is not mentioned. IR resolution is important to inform readers about the reliability of the authors' comments on IR spectra. Lack of this, Figure 3, particularly the 3a, is difficult to interpret.
- It is known that macromolecules (above ~8-10 kDa) can degrade under high energy ball milling. Although the authors compared the physical mixtures and ground materials with identical compositions, they missed inserting the native and ground PVP and HPC-SSL spectra. The overlapping signals, particularly in the case of PVP (a strong IR absorption in the region around 1650-1660 cm-1) can show similar changes in the IR spectra. Without the confirmation of the IR bands of the additives before and after the ball milling, the IR analysis of Figures 3a and 3b is weak, especially beside the unknown IR resolution.
- Although the zeta-potential differences, owing to the measurement's inherent uncertainties, are not really significant, some more details would be needed in the discussion. Additionally, although the rebamipide ZP is mentioned, ZPs of the pure (and ground) materials are missing in Table 4. Regarding the rebamipide ZP, what does "... RB was shown to be less than -20 mV ..." mean? Rather -25 mV or -15 mV? If some degradations of the additives occur during the milling, the ZP can change.
- In Figure 5, the meaning of "**" is not resolved.
Author Response
Thank you very much for reviewer’s raising important issues and providing us helpful suggestions. We have performed additional experiments and extensively revised our manuscript in the light of reviewer’s comments.
The manuscript about the nanoparticulation of rebamipide contains useful scientific information. In the current form, it requires some improvements before the appearance.
- Although the "Abstract" resolves the most used abbreviations, which is useful for those who are interested in abstract reading only, it is necessary to do also the same in the text body.
>Response
Thank you for your suggestion. We changed few words for improving the manuscript.
- In the keywords, 4 of 6 are the repetition from the title. This is unacceptable.
- Although the structure of rebamipide is shown in Figure 2, its earlier appearance, e.g., in the "Introduction" would be more advantageous.
>Response
Thank you for your suggestion. In this article we thought to describe the chemical structure formula in FTIR section. So, if it is possible, we would like to put it at this position.
- The first appearance of HPC-SSL is in line 81. The referee accepts the supplier's nickname, which appears in many publications, but it would be nice if a scientist one time gets the afford to resolve the meaning of "SSL".
>Response
Thank you for your suggestion. The name is from grade of molecular weight in HPC. So, we showed each MW in the sentences.
- Although the description of the milling machine with the name of the manufacturer is correct, it would be useful to mention that this is a mixing mill, for easier reading. It is not expectable for the reader to surf on the internet to find what the equipment nickname means.
>Response
Thank you for your suggestion. We rewrote them in revised manuscript.
- In section 2.3.2, the IR spectrometer resolution is not mentioned. IR resolution is important to inform readers about the reliability of the authors' comments on IR spectra. Lack of this, Figure 3, particularly the 3a, is difficult to interpret.
>Response
Thank you for your suggestion. We added about resolution in the method.
- It is known that macromolecules (above ~8-10 kDa) can degrade under high energy ball milling. Although the authors compared the physical mixtures and ground materials with identical compositions, they missed inserting the native and ground PVP and HPC-SSL spectra. The overlapping signals, particularly in the case of PVP (a strong IR absorption in the region around 1650-1660 cm-1) can show similar changes in the IR spectra. Without the confirmation of the IR bands of the additives before and after the ball milling, the IR analysis of Figures 3a and 3b is weak, especially beside the unknown IR resolution.
>Response
Thank you for your suggestion. We added PVP and HPC spectra in the figures and rewrote these results and discussions.
- Although the zeta-potential differences, owing to the measurement's inherent uncertainties, are not really significant, some more details would be needed in the discussion. Additionally, although the rebamipide ZP is mentioned, ZPs of the pure (and ground) materials are missing in Table 4. Regarding the rebamipide ZP, what does "... RB was shown to be less than -20 mV ..." mean? Rather -25 mV or -15 mV? If some degradations of the additives occur during the milling, the ZP can change.
>Response
Thank you for your suggestion. We rewrote in the results and discussions.
- In Figure 5, the meaning of "**" is not resolved.
>>Response
Thank you for your suggestion. We added it in this figure.

Round 2
Reviewer 2 Report
The Authors strongly improved the manuscript. In the present form, a lot of interesting data are presented and can be published in the Journal. Still, minor revision is needed. Details are listed below.
Fig. 3 - please add more space between each spectrum
Please add the histograms of particle size distribution obtained by DLS into the main text (you can choose one or more - readers usually first read the main text and in my opinion these results are interesting) - it will give them opportunities to compare with other NPs systems. Please keep in mind that one of the most important factors is which way you present the data - particle size v. number of scattered light - see recommended previously references.
Author Response
Comments of Reviewer #2
The Authors strongly improved the manuscript. In the present form, a lot of interesting data are presented and can be published in the Journal. Still, minor revision is needed. Details are listed below.
Fig. 3 - please add more space between each spectrum
Please add the histograms of particle size distribution obtained by DLS into the main text (you can choose one or more - readers usually first read the main text and in my opinion these results are interesting) - it will give them opportunities to compare with other NPs systems. Please keep in mind that one of the most important factors is which way you present the data - particle size v. number of scattered light - see recommended previously references.
>Response
Thank you for your kindly suggestion. We remade Figure 3 as your suggestion. In addition, we showed the histograms of particle size distribution as Figure 4. Then those graphs were removed from Supplemental data. Thank you very much.

Reviewer 3 Report
The authors considerably improved their manuscript. Now, this manuscript is suitable for publication in Colloids and Interfaces.
After careful reading of the current version, the referee would recommend adding the accumulation time in the particle size calculation description (line 152 of the current version, "...obtained 70 times cumulation of one sample...") if the machine software provides this data. The referee's own experiences showed some particle size dependency on the data collection period (1-5 vs. 20-30 sec).
In line 259, in the footnote of Table 3, "experiments" or "data collections" instead of "determination" would sound better.
These are tolerably minor changes and affect a couple of characters only so the authors can do them during the proofreading.
Author Response
Comments of Reviewer #3
The authors considerably improved their manuscript. Now, this manuscript is suitable for publication in Colloids and Interfaces.
After careful reading of the current version, the referee would recommend adding the accumulation time in the particle size calculation description (line 152 of the current version, "...obtained 70 times cumulation of one sample...") if the machine software provides this data. The referee's own experiences showed some particle size dependency on the data collection period (1-5 vs. 20-30 sec).
In line 259, in the footnote of Table 3, "experiments" or "data collections" instead of "determination" would sound better.
> Response
Thank you for your kindly suggestion. We added the collection period of DLS in the method section. In addition, we were able to know the scattering angle, then we also added it. And we rewrote to “experiments” in the footnote of Table 3. Thank you very much.
